# Negative Acts as Risk Factor for Work-Related Violence and Threats from Clients towards Employees: A Follow-Up Study

**DOI:** 10.3390/ijerph20043358

**Published:** 2023-02-14

**Authors:** Lars Peter Sønderbo Andersen, Karin Biering, Paul Maurice Conway

**Affiliations:** 1Danish Ramazzini Centre, Department of Occupational Medicine—University Research Clinic, Goedstrup Hospital, 7400 Herning, Denmark; 2Department of Psychology, University of Copenhagen, 1353 Copenhagen, Denmark

**Keywords:** negative acts, work-related violence, work-related threat, follow-up study

## Abstract

Background: Work-related violence and threats are major problems in many occupations, especially within the human service sector, with consequences at multiple levels, including reduced physical and mental health, increased absenteeism, and reduced organizational commitment. It is, therefore, crucial to identify risk factors for work-related violence and threats. However, only a few studies have examined whether negative acts at work increase the risk of work-related violence and threats from clients toward employees. Objective: To examine the associations between exposure to negative acts towards employees from colleagues, clients, or a combination of both, and the risk of work-related violence and threats perpetrated by clients towards employees in a longitudinal study. Methods: Questionnaire data were collected in 2010, 2011, and 2015. In total, 5333 employees from special schools, psychiatric wards, eldercare, and the Prison and Probation Services participated in the first round of data collection in 2010. Negative acts were measured in 2010 using the Short Negative Acts Questionnaire, while work-related threats and violence were measured at all three-time points. The analyses were performed using multilevel logistic regression. Results: Negative acts from clients and the combination of negative acts from both clients and colleagues were associated with later exposure to work-related violence and threats. The associations were observed after one year, and work-related threats were still present after four years. Conclusion and implications: Negative acts are associated with an increased risk of work-related violence and threats perpetrated by clients toward employees. Organizations may reduce the risk of work-related violence and threats by preventing negative acts.

## 1. Introduction

### 1.1. Negative Acts Perpetrated by Colleagues and Clients

Negative acts at work are commonly defined as “acts that are unwanted by the target that may be carried out deliberately or unconsciously but clearly cause humiliation, offence and distress” [1]. Negative acts at work can manifest in several ways, including gossiping, making remarks of a personal nature, belittling, withholding important information regarding one’s work, and excluding a colleague from the workgroup [2]. When persistently directed towards the same individual(s) and over a longer period of time, and characterized by an imbalance of power between the target and the perpetrator(s), negative acts can configure workplace bullying [3]. Negative acts and workplace bullying are also referred to as type III violence, in which a perpetrator is a current or past employee who attacks or threatens another current or past employee [4].

There is a wide variation in the prevalence of workplace bullying across the continents. Such a wide variation can be also due to workplace bullying being defined and measured in different ways [5]. A recent systematic review of the literature established a pooled prevalence of workplace bullying of 18.4% among healthcare professionals [6], with negative acts being up to sixteen times more likely to occur in the healthcare industry than in other sectors [7,8,9]. It has been argued that the higher prevalence rates of negative acts observed in the human service sectors are due to the health institutions’ hierarchical structure, multidisciplinary teams’ conflicts concerning priorities and the interpersonal and emotional nature of social and healthcare work, the hierarchical structure of healthcare institutions, and the conflicting priorities of multidisciplinary teams [10,11]. In Scandinavian countries, it is estimated that 1–5% of the workforce experiences bullying at the workplace, especially in the form of bullying behaviours perpetrated by co-workers at the same hierarchical level [11].

Even though the concept of negative acts was originally developed to signify adverse behaviours occurring between employees sharing the same workplace [2], negative acts towards employees may also be perpetrated by individuals external to the organization (not colleagues), typically service recipients (e.g., patients, relatives, clients, and pupils). For instance, a study found that in addition to colleagues, pupils and their parents can also act as perpetrators of inappropriate behaviour against teachers [12]. Other studies in the service and healthcare sectors found that not only colleagues or supervisors but also clients, patients, and their relatives could act as perpetrators of bullying [13,14,15,16]. For instance, in a study based on 14,453 participants from the general population residing in Denmark, the most commonly reported perpetrators of workplace bullying were clients (41.5%) [16]. This suggests that the phenomenon of negative acts perpetrated by service recipients is widespread.

### 1.2. Work-Related Violence and Threats

The International Labour Organization (2002) defines work-related violence as “any action, incident or behaviour that departs from reasonable conduct in which a person is assaulted, threatened, harmed, injured in the course of, or as a direct result of, his or her work” [17] (p. 4). This definition aligns with the concept of type II violence, in which a customer/client has a legitimate relationship with the workplace. Perpetrators of type II violence may thus be customers, clients, patients, students, or inmates who become violent while being served by employees at the workplace [4].

Work-related violence and threats are major issues in many occupations, being especially prevalent in the healthcare and human service sectors [18,19,20,21]. Work-related violence and threats have significant consequences at multiple levels, including physical and psychological symptoms [22,23,24], post-traumatic stress disorder [25,26], increased absenteeism, and reduced organization commitment [27,28]. Physical violence can also result in physical injuries and, in extreme cases, the death of the targeted employee [29,30]. 

Against this background, a crucial aim for both research and practice is to identify risk factors for work-related violence and threats. Work-related violence and threats are complex phenomena that are influenced by individual, structural, organizational, and cultural factors [31,32,33]. Among organizational risk factors, the quality of the psychosocial work environment plays a key role in the emergence of work-related violence and threats [18,34]. Previous studies found that high quantitative demands and work pressure, but also an adverse social environment in the form of low social support, role conflicts, and a poor social climate, are factors increasing the risk of work-related violence and threats [18,34,35]. A review found that the development of aggression in patients was related to employee dissatisfaction with the leadership and poor collaboration between nurses [36], while a study among 228 employees in six residential homes for mentally retarded patients found that a poor social climate (e.g., cliques, conflicts, or disagreements) increased the risk of work-related violence perpetrated by the patients [37]. Opposite to this, positive social relations at work can decrease the risk of work-related violence [38]. For instance, a study of 30,044 healthcare workers found that the risk of work-related violence was reduced in work units with high social capital [39]. 

Despite the critical role played by an adverse social environment, only a few studies have hitherto examined whether severe social stressors in the form of negative acts (from colleagues, clients, or both) increase the risk of subsequent exposure to work-related violence and threats perpetrated by clients towards employees. Previously, a longitudinal study found that poor interpersonal relationships between employees were associated with a higher frequency of violent acts perpetrated by patients [32]. However, the study did not measure negative acts as such, but rather the extent to which nurses experienced friendly and relaxed relations with their colleagues. Another longitudinal study did not find support for workplace bullying as a risk factor for work-related violence and threats [15]; however, this study employed a single-item measure of workplace bullying and did not differentiate between sources of bullying (i.e., perpetrated by clients or colleagues). Finally, in a cross-sectional study, negative acts perpetrated by colleagues were associated with more frequent client-perpetrated verbal violence [40]. However, the cross-sectional design makes the causal direction of the relationship observed in this study difficult to establish.

The current scarcity of longitudinal studies and the mixed findings obtained so far call for more prospective studies on the role of negative acts in the occurrence of work-related violence and threats perpetrated by clients toward employees.

### 1.3. This Present Study

In the present longitudinal study, we aimed to examine the association between negative acts perpetrated by a supervisor, subordinates, or colleagues (henceforth referred to as “colleagues”) and/or clients, pupils, patients, or inmates (henceforth referred to as “clients”) and the subsequent risk of being exposed to work-related violence and threats perpetrated by clients towards employees. More specifically, in the present study we aimed to examine the following hypothesis:Negative acts perpetrated by colleagues, clients, or the combination of both increase the risk of work-related violence and threats perpetrated by clients towards employees at the one-year and four-year follow-ups.

The longitudinal relationship between negative acts, work-related threats, and violence might be the result of stable poor working conditions over time, causing an unchanged exposure to both negative acts and work-related violence and threats. In particular, previous research suggests that both negative acts and work-related violence and threats are more likely to emerge in workplaces with a low quality of leadership [18,34,36,41,42,43,44]. We, therefore, tested the following second hypothesis:2.After adjustment for quality of leadership, negative acts perpetrated by colleagues, clients, or by the combination of both are no longer associated with an increased risk of work-related threats and violence perpetrated by clients towards employees.

## 2. Methods

### 2.1. Procedure and Participants

The study sample is based on a Danish three-wave cohort established in 2010, with follow-up measures in 2011 and 2015. Those participants not having contact with clients, having been absent from work for more than three weeks prior to receiving the questionnaire, or having been employed for less than three weeks at the workplace, were excluded from the study. Participants were recruited from the public sector, including psychiatric wards (open, acute, and secure wards), elder care, special schools (schools for pupils with special needs and diagnoses such as severe attention deficit hyperactivity disorder (ADHD) and autism), and the Prison and Probation Service (hereafter referred to as PPS). These sectors were chosen because previous research in Denmark found that they present a high prevalence of work-related violence and threats [45]. 

The participants were recruited using convenience sampling. In order to recruit participants, we first had meetings with the top managerial level, which is located in the municipalities for eldercare and special schools and in the regions for psychiatry. We then contacted the local leaders and invited the work units to participate. With regard to PPS, a meeting with the top management was arranged, followed by an invitation to participate sent to all employees. The recruitment and data collection procedures are described in more detail elsewhere [46].

In 2010 and 2011, participants filled out a paper-and-pencil questionnaire during a planned meeting at the worksite. The invited participants had to be employed in jobs involving contact with clients, and they must not have had more than three weeks of absence prior to the survey. Employees who did not participate in the meeting received a questionnaire in a prepaid stamped envelope to be returned directly to the researchers. For employees working in the PPS, a web-based questionnaire was used following a decision taken by the top management. It was clearly stated in the cover letter of the questionnaires that participation in the study was voluntary and that the data would be treated confidentially. In 2015, using civil registration numbers, we were able to contact the participants from 2010 at their home addresses, regardless of whether they were still employed in their original workplace. 

The initial cohort (2010) consisted of 5333 persons: psychiatry (*N* = 909/response rate: 85%), special schools (*N* = 731/response rate: 91%), elder care (*N* = 940/response rate: 80%), and PPS (*N* = 2753/response rate: 60%). Eligible for taking part in the follow-up in 2011 were participants who answered the questionnaire in 2010, worked at the same work unit as in 2010, and had no more than three weeks of absence at the time of the second survey in 2011. Between the 2010 and 2011 measurements, a few employees changed jobs, retired, or were on sick leave or maternity leave. Altogether, 3584 participants answered both the 2010 questionnaire and the questionnaire in 2011. In the third wave of data collection in 2015, all participants from 2010 were invited to participate in a new round. In all, 3486 answered the questionnaire in both 2010 and 2015. We used civil registration numbers to match every participant on an individual level. 

Since the risk of work-related violence and threats, negative acts, and the quality of leadership may vary from one workplace to another, in this study we only included employees working at the same workplace in 2010, 2011, and 2015. We thus excluded 805 participants who reported that they were not working any longer in the same workplace in 2015 as in 2010 and 2011.

### 2.2. Questionnaires

#### 2.2.1. Exposure: Negative Acts at Work

We measured negative acts in 2010 by using the short version of the Negative Acts Questionnaire (SNAQ) [47], which includes nine items capturing a series of negative and unwanted behaviours, both direct (e.g., openly attacking the victim) and indirect (e.g., social isolation and slander). The respondents were asked to report how often in the preceding six months they had been exposed, at their present workplace, to each of the nine negative acts, with response options ranging from 1 to 5 (“never”, “now and then”, “monthly”, “weekly”, and “daily”). We computed an overall sum scale (continuous scores) including all nine items. Missing items were equated to never (1). The possible range of the sum score was 9–45, therefore, we subtracted 9 to obtain a range from 0 to 34. The sum scores were then dichotomized into two categories consisting of those exposed to negative acts regardless of frequency (i.e., sum scores above 0) and a reference group including those not exposed to negative acts (i.e., sum scores equal to 0). In case respondents reported being exposed to at least one negative act, they were additionally asked to indicate the source of the negative act by the following question: “If yes, from whom? Colleagues; Manager/supervisor; Subordinates; Clients/patients/pupils/inmates; Relatives”. Respondents were allowed to report more than one source. The Cronbach’s alpha for the Negative Acts Scale was 0.85. The SNAQ has been found to have good psychometric properties [48].

#### 2.2.2. Outcome: Work-Related Violence and Threats

We applied a broad definition of work-related violence that included threats of violence as well as actual physical violence [49]. We used a checklist consisting of 11 different types of violent incidents and 7 different types of threats of violence employed in previous research in Sweden [50]. Threatening behaviours included being threatened with beatings, written threats, threatened in a scolding manner, threatened in an insulting manner, threatened over the phone, threatened with objects, and threatened indirectly (threats towards employees’ family members). Respondents were asked to indicate how often during the past year they had experienced each of these different types of threats or violence at the workplace. The forms of physically violent behaviour examined were being hit, spit on, hit with an object, scratched/pinched, shoved, held firmly, punched with a fist, kicked, bit, having a hard object thrown at you, and use of a weapon or a weapon-like object. Further description of the items can be seen in Rasmussen et al. (2013) [46]. For both work-related threats and violence, the frequency of occurrence was measured with a five-point Likert-like scale ranging from never (0) to almost daily (4). The items were computed into two separate sum scales labelled “threats of violence” and “physical violence”. Missing items were coded as never (0). Therefore, the sum scores for the threats scale could range from 0 to 28, while the sum scores for the violence scale could range from 0 to 44. The sum scores were then dichotomized into two categories consisting of those exposed to work-related threats (or violence), regardless of frequency (i.e., sum scores above 0), and a reference group including those not exposed to work-related threats (or violence; i.e., sum scores equal to 0). Respondents reporting work-related threats or violence were also asked the following question: “If yes, from whom? Colleagues; Manager/supervisor; Subordinates; Clients/patients/pupils/inmates”. Respondents were allowed to report more than one source. Cronbach’s alphas were 0.87 and 0.81 for the work-related violence scale and the work-related threats scale, respectively. The Cronbach’s alpha for the threat scale was 0.81 and the Cronbach’s alpha for the violence scale was 0.90.

### 2.3. Covariates

The analyses were adjusted for gender, age, and quality of leadership. Gender and age were derived from the civil registration numbers. 

Quality of leadership was measured with a scale from the second version of the Copenhagen Psychosocial Questionnaire (COPSOQ) [51]. The scale consisted of four questions (for instance, “To what extent would you say that your immediate superior is good at work planning?”; “To what extent would you say that your immediate superior is good at solving conflicts?”). The items are measured using a five-point Likert scale ranging from “To a very low extent” (0) to “To a very large extent” (100). Scale scores ranged from 0 to 100, with higher scores indicating higher levels of quality of leadership. Cronbach alpha for the quality of leadership scale was 0.84. The COPSOQ offers reliable and distinct measures of a wide range of psychosocial dimensions of modern working life [52].

### 2.4. Other Variables

Information about whether participants in 2015 were working at the same workplace as in 2010 and 2011 was obtained in the 2015 survey by a single item. Information about baseline violence and threats were obtained from the 2010 questionnaire. The violent and threats items were identical to the 2011 violent and threat items (see outcome variables). 

### 2.5. Statistical Analyses

First, we created four categories of perpetrators of negative acts, namely “colleagues” (including co-workers, supervisors, and subordinates), “clients” (including clients, patients, pupils, inmates, and their relatives (very few)), “both colleagues and clients”, which include those cases where respondents reported that they had been exposed to negative acts by the combination of both clients and colleagues, and finally “unknown perpetrator”, which includes those respondents who did not report the source of exposure. The main analyses were performed among all employees working at the same workplace at the follow-up (in 2011 and 2015) as in 2010. The associations were examined using multilevel logistic regressions, to take into account that the answers provided by employees from the same workplace might not be independent. In the first step of the regression (model 1, unadjusted), we estimated the crude odds ratios (OR), together with their respective 95% confidence intervals (95% CI), for the separate associations of each of the four categories of exposure measured in 2010 with work-related violence and threats measured in 2011 and 2015. In the second step (model 2), we repeated the same analyses while adjusting for gender, age, and baseline work-related threats and violence. In the third and last step (model 3), the quality of leadership measured in 2010 was entered as an additional confounder. All statistical analyses were performed using SPSS (Version 20) (IBM Corp, Armonk, NY, USA).

## 3. Results

The sample descriptives are presented in Table 1. Most participants were women and older than 40 years. In 2010, 24.8% of the participants reported exposure to negative acts from colleagues, 18.1% from clients, and 19.5% from both. In 2011, 62.9% and 35.7% of the participants reported exposure to work-related threats and violence, respectively, at least once during the last year. In 2015, 65.7% and 39% of the participants reported exposure to work-related threats and violence, respectively, at least once during the last year.

### 3.1. Associations between Negative Acts and Work-Related Threats

Table 2 shows the associations between negative acts from colleagues, clients, the combination of both colleagues and clients, and from an unknown perpetrator as measured in 2010, and work-related threats perpetrated by clients towards employees as measured in 2011 and 2015. In the unadjusted model (Model 1), negative acts perpetrated by clients and negative acts perpetrated by both colleagues and clients were associated with work-related threats at both the one-year and the four-year follow-up. These associations also remained significant in Model 2, which was adjusted for gender, age, and previous threats. In the final model (Model 3), adjusting for the quality of leadership did not change the associations, although the latter became somewhat stronger in the four-year follow-up.

### 3.2. Associations between Negative Acts and Work-Related Violence

Table 3 shows the associations between negative acts from colleagues, clients, or from both colleagues and clients, and from an unknown perpetrator as measured in 2010, and work-related violence perpetrated by clients towards employees as measured in 2011 and 2015. In the one-year follow-up, negative acts perpetrated by clients and by the combination of both colleagues and clients were significantly associated with work-related violence in both the unadjusted and adjusted models. Concerning the four-year follow-up, no significant associations were found in the adjusted models. 

## 4. Discussion

In the present study, we aimed to examine the one- and four-year associations between negative acts perpetrated by colleagues, clients, or a combination of both and the risk of subsequent exposure to work-related threats and violence perpetrated by clients toward employees. We found that negative acts perpetrated by a combination of colleagues and clients were associated with an increased risk of work-related threats in both the one-year and the four-year follow-up, as well as with an increased risk of work-related violence, but only in the one-year follow-up.

### 4.1. Strengths and Limitations

Major strengths of the present study are the adoption of a longitudinal design, the high response rates, which are above average for organizational surveys [53], and the inclusion of participants from four sectors in Denmark, all of which are characterized by a high risk of work-related threats and violence. All these elements increase the generalizability of our findings to similar workplaces. It is a further strength that we included employees who were working at the same work unit at all follow-up measurement points. However, this may have introduced a healthy worker effect bias if those more exposed to negative acts or threats/violence had a higher risk of leaving their workplace, which would result in a possible underestimation of the associations. Furthermore, we applied multilevel analysis taking into account that the answers provided by employees from the same workplace might not be independent.

Despite these strengths, the results of the present study should be interpreted in light of a number of possible limitations. First, the participants were not necessarily representative of their respective occupational sectors, as they were recruited using a convenience sampling design. A few workplaces refused to participate, and selection bias could not be ruled out. However, data from the PPS were collected from the whole sector. Second, the method for data collection employed in the PPS differed from that used in the other three sectors. This may have affected the response rate, which was lower in the PPS, potentially introducing an additional source of selection bias. Third, the data were entirely based on self-reported measures, which may have caused common method variance, potentially inflating the associations between the examined variables [54]. Common method variance is reduced, however, when using longitudinal analyses. Fourth, the six-month and one-year windows we employed when measuring the exposure and the outcome, respectively, might have introduced recall bias. Recall time windows of these lengths are commonly employed in the field of work-related violence and workplace bullying. To overcome recall bias, weekly or daily measurements are necessary [55]. However, these may induce other problems in the form of lower response rates and response shifts [56]. Fifth, in all, 8% were employed for less than one year at their current workplace. Therefore, only a few participants were employed for less than 6 months at the current workplace (corresponding to the recall period of the S-NAQ) and may have been exposed to negative acts at a workplace other than the current one. Another limitation is related to the possibility that negative acts at work and work-related threats and violence do not represent distinct constructs. In a supplementary analysis, we found that, in the 2010 wave, scores on the NAQ and scores on the scale used to measure work-related threats and violence were weakly correlated (r = 0.26 and r = 0.09, respectively), which suggests that negative acts and work-related violence/threats are separate constructs. The distinction between the two constructs is, however, supported in previous research [57], and furthermore, the longitudinal design enabled us to temporally separate measures of the two constructs. A further limitation is the use of dichotomized measures, which might result in a loss of information and a simplification of the relationships under scrutiny. Therefore, we performed a supplementary analysis employing the SNAQ as a continuous score in the final model (model 3). The associations point in the same direction compared to the analysis based on SNAQ as a dichotomized variable (data not shown). Despite their limitations, dichotomizing variables might also have advantages as estimates such as odds ratios may provide results that are more easily understandable for a wider audience [58]. Furthermore, the dichotomization of negative acts might support the identification of employees who are particularly at risk for work-related threats and violence. We report non-statistical associations because recent recommendations encourage researchers not to draw conclusions based solely on statistical significance but rather to consider the size and direction of the estimates [59] since the non-significant results could be due to a possible type 2 error.

### 4.2. Interpretation

Supporting the first study hypothesis, one possible explanation for the observed associations might be that being exposed to negative acts is an established risk factor for mental health problems [6,60,61,62], irrespective of the perpetrator (clients or colleagues) [16]. Reduced mental health might then result in deteriorated social interactions, making employees more vulnerable to client-related aggression and assaults [63,64,65]. Corroborating this, a few previous studies suggest that lower levels of mental health may influence the quality of interactions between service providers and service recipients, resulting in associations whereby mental health problems contribute to work-related violence [63,66]. For instance, in a two-year prospective study, employees with sleep disturbances were found to have a higher risk of being exposed to work-related violence [67]. 

The fact that in previous studies, mental health problems following negative acts were shown to last for up to three to five years [60,62] can explain why negative acts may represent a risk factor for work-related threats in the one- and four-year intervals adopted in the present study. In sum, the results of this study may point to some stability over time due to the result that the association between negative acts and work-related threats was significant in both the one-year and four-year follow-ups. 

In rejection of the second hypothesis, we found that the associations between negative acts and the risk of work-related threats and violence were not affected when adjusting for the quality of leadership. Even though previous studies found that quality of leadership influences both negative acts as well as work-related violence and threats [18,34,36,41,42,43,44], in the present study quality of leadership did not influence the association between negative acts and work-related threats and violence. One explanation may be that previous studies that found associations between leadership and negative acts were conducted in occupations that differ from the ones included in the present study. For instance, Nielsen et al.’s (2015) study focused on seafarers from two Norwegian shipping companies [43], while the study of Ballien et al. (2014) was carried out among employees working in the textile industry and financial services in Belgium [42]. As mentioned earlier, higher prevalence rates of negative acts have been observed, especially in the human service sectors, among other reasons, due to the multidisciplinary teams’ conflicting priorities and healthcare institutions’ hierarchical structure [10,11] thus, the structure and conflicting priorities in a higher degree may influence the risk for negative acts than the quality of leadership.

### 4.3. Generalizability

The study is large and based on many workplaces in the human service sector, and furthermore, the response rate is high. Therefore, the results of this study may be transferred to similar human service sector workplaces when taking the above-mentioned limitations into account.

## 5. Conclusions

The present study suggests that negative acts from clients and the combination of negative acts from both clients and colleagues play a role in increasing the risk of later work-related threats and violence from clients toward employees. 

This finding indicates that negative acts could be targeted in an effort to prevent work-related threats and violence in human service settings. Previous studies suggest that the prevention of negative acts may be addressed through the implementation of organizational policies, practices, and procedures; in particular, organizations should commit themselves to designing specific policies targeting such behaviours [68,69], including zero-tolerance policies for threats and violence such as those already implemented in a number of workplaces [70]. We recommend organizations also include the prevention of negative acts in their efforts to reduce the risk of work-related violence and threats.

## Figures and Tables

**Table 1 ijerph-20-03358-t001:** Descriptive data of participants in 2010, 2011, and 2015.

		2010 (*n* = 5333)		2011 (*n* = 3751)		2015 (*n* = 3486)	
		*n*	%	*n*	%	*n*	%
Age group	<40	3262	61.2	2413	64	2285	65
	≥40	2071	38.8	1338	36	1201	35
Sex	Female	3429	64.3	2476	66	2258	65
	Male	1904	35.7	1275	34	1228	35
In 2015 working at the same work unit as in 2010 and 2011	Yes					2152	72.8
	No					805	27.2
Exposed to negative acts during the past six months 2010	Never	1413	26.5				
Exposed to negative acts from colleagues during the past six months 2010	One time or more	1325	24.8				
Exposed to negative acts from clients during past six months 2010	One time or more	967	18.1				
Exposed to negative acts from clients and colleagues during past six months 2010	One time or more	1041	19.5				
Exposed to negative from unknown source	One time or more	587	11.0				
Exposed to threats during past year 2010	Never	1866	35.4				
	One time or more	3404	64.6				
Exposed to violence during past year 2010	Never	3195	61.3				
	One time or more	2020	38.7				
Exposed to threats during past year 2011	Never			1337	37.1		
	One time or more			2270	62.9		
Exposed to violence during past year 2011	Never			2307	64.4		
	One time or more			1275	35.6		
Exposed to threats during past year 2015	Never					1006	34.3
	One time or more					1927	65.7
Exposed to violence during past year 2015	Never					1796	61.0
	One time or more					1148	39.0

**Table 2 ijerph-20-03358-t002:** Associations between negative acts perpetrated by colleagues, clients, or the combination of both colleagues and clients and work-related threats.

	2011			
	Unadjusted	Model 1	Model 2	Model 3
Non-exposed (ref)	1	1	1	1
Colleagues	1.18 (0.93–1.51)	1.16 (0.91–1.49)	1.07 (0.81–1.43)	0.97 (0.72–1.31)
Clients	2.93 (2.24–3.83) **	2.76 (2.10–3.63) **	1.53 (1.13–2.09) **	1.42 (1.03–1.96) *
Colleagues and clients	4.09 (3.10–5.41) **	3.72 (2.80–4.98) **	1.98 (1.44–2.71) **	1.73 (1.24–2.41) **
None specified	0.85 (0.66–1.09)	0.86 (0.67–1.10)	0.94 (0.68–1.20)	0.87 (0.65–1.17)
	**2015**			
	**Unadjusted**	**Model 1**	**Model 2**	**Model 3**
Non-exposed (ref)	1	1	1	1
Colleagues	1.36 (0.92–1.99)	1.44 (0.96–2.15)	1.32 (0.87–2.02)	1.45 (0.92–2.32)
Clients	3.04 (2.18–5.30) **	3.03 (1.92–4.80) **	2.04 (1.25–3.36) **	2.37 (1.41–4.00) *
Colleagues and clients	3.60 (2.35–5.52) **	3.44 (2.20–5.34) **	1.95 (1.23–3.14) **	2.04 (1.22–3.41) *
None specified	0.87 (0.59–1.30)	0.98 (0.65–1.49)	1.01 (0.65–1.57)	1.13 (0.69–1.82)

** *p* < 0.01; * *p* < 0.05; Model 1 adjusted for gender and age; Model 2 adjusted for gender, age, and baseline threats; Model 3 adjusted for gender, age, baseline threats, and quality of leadership; Colleagues: Coworkers, supervisors, and subordinates; Clients: Patients, elderly, clients, pupils, and inmates; None specified: No information about perpetrator.

**Table 3 ijerph-20-03358-t003:** Associations between negative acts perpetrated by colleagues, clients, or the combination of both colleagues and clients and work-related violence.

	2011			
	Unadjusted	Model 1	Model 2	Model 3
Non-exposed (ref)	1	1	1	1
Colleagues	1.37 (1.03–1.81) *	1.27 (0.96–1.68)	1.15 (0.82–1.61)	1.10 (0.77–1.58)
Clients	2.32 (1.70–3.15) *	2.15 (1.62–2.86) **	1.56 (1.12–2.25) *	1.50 (1.03–2.16) *
Colleagues and clients	2.66 (2.02–3.50) *	2.44 (1.84–3.24) **	1.70 (1.20–2.04) *	1.62 (1.13–2.33) *
None specified	1.21 (0.93–1.57)	1.13 (0.85–1.50)	1.23 (0.87–1.74)	1.25 (0.87–1.79)
	**2015**			
	**Unadjusted**	**Model 1**	**Model 2**	**Model 3**
Non-exposed (ref)	1	1	1	1
Colleagues	1.15 (0.74–1.80)	1.24 (0.78–1.96)	1.12 (0.69–1.82)	1.12 (0.66–1.91)
Clients	2.02 (1.29–3.18) *	1.79 (1.13–2.84) *	1.52 (0.93–2.49)	1.56 (0.93–2.68)
Colleagues and clients	2.06 (1.34–3.22) *	1.99 (1.27–3.13) *	1.36 (0.84–2.21)	1.26 (0.74–2.14)
None specified	0.99 (0.63–1.60)	1.15 (0.71–1.86)	1.17 (0.70–1.95)	1.06 (0.61–1.85)

** *p* < 0.01; * *p* < 0.05; Model 1 adjusted for gender and age; Model 2 adjusted for gender, age, and baseline threats; Model 3 adjusted for gender, age, baseline threats, and quality of leadership; Colleagues: Coworkers, supervisors, and subordinates; Clients: Patients, elderly, clients, pupils, and inmates; None specified: No information about perpetrator.

## Data Availability

The study was approved by the Danish Data Protection Agency. According to Danish Law, Act on Research Ethics Review of Health Research Projects, (available at: www.nvk.dk/english/act-on-research, accessed on 9 February 2023). The datasets used and/or ana-lysed during the current study are available from the authors on reasonable request.

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
