# Peer review of "Negative Acts as Risk Factor for Work-Related Violence and Threats from Clients towards Employees: A Follow-Up Study"

_ijerph, 2023, doi:10.3390/ijerph20043358_

Round 1

Reviewer 1 Report

Thank you for your effort to share and publish this article. However, there are several comments that need further clarification and improvement.

1. To paraphrase several sentences to secure your original works. Reduce to less than 20% (Turnitin is kindly referred 31% similarity index with excluded references ).

2. To add literature reviews last 5 to 10 years back (Introduction-1.1. Negative acts perpetrated by colleagues and clients).

3. Methods:

a. Kindly report in the (Result part-Descriptive data of participants based on your final samples NOT your original samples (N=5333). How many of your final samples. Report each study phases results/outcomes based on your final samples. It is enough to report your original samples (N=5333) in the (Method-Procedure and participants).

b. Questionnaires (Exposure: Negative acts at work & Outcome: Work-related violence and threats) to measure two constructs of your study are little bit confusing. You also acknowledged them as a limitation in your study. Therefore:

i. Could you provide the questionnaires for further perusal

ii. Could you provide the discriminant index value

If both scales measure the same construct, if possible, I would suggest to examine based on emotion reaction e.g anger or verbal violence toward physical violence as its impacts.

4. As the scales used to measure constructs are my major concerns. The other rest of your writing is affected accordingly particularly the discussion part.

Author Response

See attashed file

Reviewer 2 Report

This is an interesting paper and the overall quality of the study is good. Method is generally sound and limitations are highlighted in discussion. I don’t have much to comment on besides a few minor ones:

General:

Typo – line 338 (negative acts), line 342 (Nielsen et al), line 344 (Baillien et al)

Method:

For the inclusion criteria, workers recruited were working for at least how long? The SNAQ ask about negative acts exposed to within the past six months – thus workers recruited should have worked at their workplace for at least 6 months. However, this was not clear from the inclusion criteria described.

Though at present there is no single, consistent definition of workplace violence and universal method for measuring it – the psychometric properties of the tool used to measure it ensures that the results are credible. However, there was no mention of the validity for SNAQ, workplace violence checklist, and COPSOQ – the authors only provided internal consistency reliability indices (Cronbach’s alpha).

All the best.

Author Response

Reviewer 2

Dear reviewer

Thank you for your constructive comments.

We believe that we have payed careful attention to all your comments and have addressed each comment. We have developed the paper, and we think that the manuscript has been improved and we hope you will find the paper suitable for the audience of the journal.

Typo – line 338 (negative acts), line 342 (Nielsen et al), line 344 (Baillien et al)

Authors’ response: The typos have been corrected

 Method:

For the inclusion criteria, workers recruited were working for at least how long? The SNAQ ask about negative acts exposed to within the past six months – thus workers recruited should have worked at their workplace for at least 6 months. However, this was not clear from the inclusion c riteria described.

Authors’ response: Thank you for your constructive comments. We added more information about inclusion criteria. Specifically, we added the following (page 4): “Those participants not having contacts with clients, having being absent from work for more than 3 weeks prior to receiving the questionnaire, or having been employed for less than 3 weeks at the workplace were excluded from the study.

Further, we added the following in the limitation section (page 10): “In all, 8 % were employed for less than one year at the current workplace. Therefore, only a few participants were employed for less than 6 months at the current workplace (corresponding to the recall period of the S-NAQ) and may have been exposed to negative acts at another workplace than the current one. 

Though at present there is no single, consistent definition of workplace violence and universal method for measuring it – the psychometric properties of the tool used to measure it ensures that the results are credible. However, there was no mention of the validity for SNAQ, workplace violence checklist, and COPSOQ – the authors only provided internal consistency reliability indices (Cronbach’s alpha).

Authors’ response: Thank you for this comment. We added Cronbach’s alphas as requested (page 5). We have added this: The cronbach’s alpha for the threats scale was 0.81 and the cronbach’s alpha for the violence scale was 0.90

Furthermore, we have added this (page 5): The COPSOQ offers reliable and distinct measures of a wide range of psychosocial dimensions of modern working life (Burr, Berthelsen et al. 2019).

And finally, we have added this (page 5): The SNAQ has been found to have good psychometric properties (Kiprillis, Gray et al. 2022)

Reviewer 3 Report

Dear Authors

This is an apt research theme and the manuscript has been well crafted.

There are some minor editorial changes required such as inappropriate brackets and other punctuation marks.

Wishing you all the best.

Reviewer

Author Response

Reviewer 3

This is an apt research theme and the manuscript has been well crafted.

There are some minor editorial changes required such as inappropriate brackets and other punctuation marks.

Authors’ response: We have checked the manuscript

Wishing you all the best.

Authors’ response: Thank you.

Reviewer 4 Report

General comments: An interesting study that investigates some of the less explored relations between different sorts of negative acts in the workplace. It contributes to the occupational bullying research field and the occupational violence research field. A well-written paper. I have only few comments and suggestions.

I have one significant objection: The study’s results support only an association between negative acts from clients and subsequent violence/threats from clients. The results do not support an association between negative acts from colleagues and subsequent violence/threats. When the negative acts come solely from colleagues the results (Table 2 and 3) clearly shows that the association with violence/threats is statistically non-significant. I suggest that the authors amend the text to reflect that clearly, including in abstract and in the conclusion. The authors' conclusion (about the role of the colleagues) are incorrect for the following reason:

First, the only real information about the role of the type of perpetrator (colleagues or clients) comes from analyses of the two exposure groups “colleagues” and “clients”, where respondents report exposure to negative acts from just one of the perpetrator types, but not the other. The exposure group “colleagues and clients” is (almost) irrelevant in this respect, because it is impossible to draw conclusions as to which perpetrator group is important. It could be one of them, or both.

Thus, a sentence such as “…Negative acts from clients and colleagues…were associated with later exposure to work-related violence and threats” (abstract. L 22-23) is misleading. While it is true in the sense that the statistical estimate was significant for the “colleagues and clients” exposure group, it is false because statistical significance in this case does not tell which of the perpetrator types that play a role.

More examples:

“…negative acts perpetrated by colleagues and clients were associated with an increased risk…” (L 282): This is also misleading, for the same reasons as above.

“…The present study suggest that negative acts from colleagues and clients, as well as from both, play a role in increasing the risk...” (L 356-7): Again I would say no. As just explained, the analysis of the exposure group “colleagues and clients” tells nothing of which type of perpetrator that is important.

For the same reason there is no basis to recommend that that policies addressing threats and violence include negative acts perpetrated by employees towards colleagues (L. 365-6).

Less important comments:

A wide range of exposures to negative acts (scores from 1-34) are collapsed to just a single outcome: “Exposed”. It would strengthen the results if a dose-response relationship between the exposure and outcome was investigated.

Table 1: A minor typo (“colloagues”).

Author Response

Reviewer 4

Dear reviewer

Thank you for your constructive comments.

We believe that we have payed careful attention to all your comments and have addressed each comment. We have developed the paper, and we think that the manuscript has been improved and we hope you will find the paper suitable for the audience of the journal.

Comments and Suggestions for Authors

General comments: An interesting study that investigates some of the less explored relations between different sorts of negative acts in the workplace. It contributes to the occupational bullying research field and the occupational violence research field. A well-written paper. I have only few comments and suggestions.

Authors’ response: Thank you very much.

I have one significant objection: The study’s results support only an association between negative acts from clients and subsequent violence/threats from clients. The results do not support an association between negative acts from colleagues and subsequent violence/threats. When the negative acts come solely from colleagues the results (Table 2 and 3) clearly shows that the association with violence/threats is statistically non-significant. I suggest that the authors amend the text to reflect that clearly, including in abstract and in the conclusion. The authors' conclusion (about the role of the colleagues) are incorrect for the following reason:

Authors’ response: Thank you for this constructive comment. Recent recommendations encourage researchers not to draw conclusions based solely on statistical significance, but rather to consider the size and direction of the estimates (Wasserstein & Lazar, 2016), since the non-significant results could be due to possible type-2 error.

If we look at the size and direction of the estimates concerning the association between negative acts perpetrated by co-workers and risk for work-related violence perpetrated by clients, the estimates all points to an increased risk, although the associations are not statistically significant. Based on these recommendations, we do not believe that our conclusions are incorrect However, after a new look on the tables, we agree that the OR are very close to 1. We have reduced the focus on colleagues, checked the manuscript for this and changed the wording in the conclusion.

We have changed the beginning of the conclusion to this: The present study suggests that negative acts from clients and the combination of both clients and colleagues play a role in increasing the risk of later work-related threats and violence from clients towards employees.

We added the following in the limitation section (page 10): “Recent recommendations encourage researchers not to draw conclusions based solely on statistical significance, but rather consider the size and direction of the estimates (Wasserstein & Lazar, 2016), since the non-significant results could be due to possible type-2 error.”

First, the only real information about the role of the type of perpetrator (colleagues or clients) comes from analyses of the two exposure groups “colleagues” and “clients”, where respondents report exposure to negative acts from just one of the perpetrator types, but not the other. The exposure group “colleagues and clients” is (almost) irrelevant in this respect, because it is impossible to draw conclusions as to which perpetrator group is important. It could be one of them, or both.

Thus, a sentence such as “…Negative acts from clients and colleagues…were associated with later exposure to work-related violence and threats” (abstract. L 22-23) is misleading. While it is true in the sense that the statistical estimate was significant for the “colleagues and clients” exposure group, it is false because statistical significance in this case does not tell which of the perpetrator types that play a role.

We have changed the in the abstract: Negative acts from clients and the combination of both clients and colleagues…..

More examples:

“…negative acts perpetrated by colleagues and clients were associated with an increased risk…” (L 282): This is also misleading, for the same reasons as above.

“…The present study suggest that negative acts from colleagues and clients, as well as from both, play a role in increasing the risk...” (L 356-7): Again I would say no. As just explained, the analysis of the exposure group “colleagues and clients” tells nothing of which type of perpetrator that is important.

For the same reason there is no basis to recommend that that policies addressing threats and violence include negative acts perpetrated by employees towards colleagues (L. 365-6).

Authors’ response: Thank you for this constructive comments concerning the exposure group “colleagues and clients”. We disagree that this group is irrelevant because it consists of both sources. In the methods section, we described the procedure for filling out the SNAQ: "In case respondents reported being exposed to at least one negative act, they were additionally asked to indicate the source of the negative act by the following question: “If yes, from whom? Colleagues; Manager/supervisor; Subordinates; Clients/patients/pupils/inmates; Relatives. Respondents were allowed to report more than one source".

The group "colleagues and clients" thus consists of those cases where respondents report that they have by exposed to negative acts by both clients and colleagues. We agree we cannot identify which source was responsible for the association; however, we distinguished the perpetrator in the other analysis. 

We have added this in the statistical analyses section (page 7)….both colleagues and clients, "which include those cases where respondents report that they have by exposed to negative acts by both clients and colleagues…

Less important comments:

A wide range of exposures to negative acts (scores from 1-34) are collapsed to just a single outcome: “Exposed”. It would strengthen the results if a dose-response relationship between the exposure and outcome was investigated.

Our reply: Thank you for this relevant comment. We made a supplementary analysis applying SNAQ as a continuous score in the final model (model 3). The results point in the same direction compared to the analysis based on SNAQ as dichotomized variable.

We added the following in the revised version, limitation section (page 10): "A further limitation is the use of dichotomized measures, which might result in a loss of information and a simplification of the relationships under scrutiny. Therefore, we performed a supplementary analysis employing the SNAQ as a continuous score in the final model (model 3). The associations point in the same direction compared to the analysis based on SNAQ as dichotomized variable (data not shown). Despite its limitations, dichotomizing variables might also have advantages as estimates such as odds ratios may provide results that are more easily understandable for a wider audience. Furthermore, the dichotomization of negative acts might support the identification of employees who are particularly at risk for work-related threats and violence.

Table 1: A minor typo (“colloagues”).

Reply: We corrected this typo.

Round 2

Reviewer 4 Report

Thanks for amending and improving the manuscript. I still don't agree that the results support a recommendation that policies should include negative acts perpetrated by employees towards colleagues. As you rightly pointed out in your response to my criticism, the OR of the group 'Colleagues' is very close to 1 (1.10-1.15 in model 2 and 3) and with broad confidence intervals. Therefore, workplaces risk spending valuable resources following a recommendation that is based on the interpretation that statistically non-significant OR's around 1.10-1.15 shows a trend.

Author Response

Authors' response: Thank you for your comment. We have changed the final sentence to this: We recommend organizations also to include the prevention of negative acts in their efforts to reduce the risk of work-related violence and threats.

 We hope that this revision will meet your concerns about the previous recommendation
